# Antiadhesive Hyaluronic Acid-Based Wound Dressings Promote Wound Healing by Preventing Re-Injury: An In Vivo Investigation

**DOI:** 10.3390/biomedicines12030510

**Published:** 2024-02-23

**Authors:** Da Som Kim, Keum-Yong Seong, Hyeseon Lee, Min Jae Kim, Sung-Min An, Jea Sic Jeong, So Young Kim, Hyeon-Gu Kang, Sangsoo Jang, Dae-Youn Hwang, Sung-Baek Seo, Seong-Min Jo, Seung Yun Yang, Beum-Soo An

**Affiliations:** 1Department of Biomaterials Science (BK21 FOUR Program), College of Natural Resources and Life Science, Life and Industry Convergence Research Institute, Pusan National University, Miryang 50463, Republic of Korea; kds4505@pusan.ac.kr (D.S.K.); ky.seong0124@gmail.com (K.-Y.S.); hyeslee96@gmail.com (H.L.); kmjkmj0804@naver.com (M.J.K.); jjsic123@gmail.com (J.S.J.); thdud1621@naver.com (S.Y.K.); mukal354@naver.com (H.-G.K.); 14sangsoo@gmail.com (S.J.); dyhwang@pusan.ac.kr (D.-Y.H.); sbseo81@pusan.ac.kr (S.-B.S.); seongmini@pusan.ac.kr (S.-M.J.); 2Division of Endocrinology, Department of Internal Medicine, University of California Davis School of Medicine, Davis, CA 95817, USA; an20381567@gmail.com

**Keywords:** hyaluronic acid, wound dressing, foam dressing, biocompatibility, re-injury

## Abstract

Wound dressings are widely used to protect wounds and promote healing. The water absorption and antifriction properties of dressings are important for regulating the moisture balance and reducing secondary damages during dressing changes. Herein, we developed a hyaluronic acid (HA)-based foam dressing prepared via the lyophilization of photocrosslinked HA hydrogels with high water absorption and antiadhesion properties. To fabricate the HA-based foam dressing (HA foam), the hydroxyl groups of the HA were modified with methacrylate groups, enabling rapid photocuring. The resulting photocured HA solution was freeze-dried to form a porous structure, enhancing its exudate absorption capacity. Compared with conventional biopolymer-based foam dressings, this HA foam exhibited superior water absorption and antifriction properties. To assess the wound-healing potential of HA foam, animal experiments involving SD rats were conducted. Full-thickness defects measuring 2 × 2 cm^2^ were created on the skin of 36 rats, divided into four groups with 9 individuals each. The groups were treated with gauze, HA foam, CollaDerm^®^, and CollaHeal^®^ Plus, respectively. The rats were closely monitored for a period of 24 days. In vivo testing demonstrated that the HA foam facilitated wound healing without causing inflammatory reactions and minimized secondary damages during dressing changes. This research presents a promising biocompatible foam wound dressing based on modified HA, which offers enhanced wound-healing capabilities and improved patient comfort and addresses the challenges associated with conventional dressings.

## 1. Introduction

Skin serves several crucial roles and functions as a natural barrier that shields the body from harmful environmental substances [1]. Wounds are defined as the disruption of the continuity of a body structure, including skin, tissues, and mucous membranes, resulting from physical or thermal damages. They are classified as acute wounds or chronic wounds based on the duration and characteristics of the healing process [2]. Acute wounds, such as skin wounds or surgical wounds, typically heal completely within a few weeks through the natural wound-healing process, depending on the size and depth of the wounds. Chronic wounds, including pressure ulcers, leg ulcers, severe burns, and diabetic ulcers, are more difficult to control because of their slow healing time, persistence, and abnormal healing process. Thus, treating chronic wounds as a long-lasting problem is important because the number of patients with chronic wounds is expected to increase with the aging population [2].

General wound healing involves a sequential series of events, including inflammations, a formation of granulation tissues, matrix remodeling, and re-epithelialization [3]. Exposure of a wound to the external environment can lead to infections and drying out, posing challenges in its treatment. Dry wound dressings, such as gauze and bandages, are commonly employed during the initial phases of wound healing [4]. Nonetheless, these dry dressings are difficult to provide a moist healing environment, which is essential for retaining wound exudates and promoting effective wound healing. Furthermore, they tend to adhere to wounds, leading to complications such as wound rupture, inflammation, and pain upon removal. Consequently, the newly formed skin or tissue can be compromised, potentially causing re-injury and bleeding [5,6].

Several types of dressings provide an ideal environment for temperature and humidity to stimulate wound healing. They also have non-adhesive properties to minimize the pain during dressing changes for patients, overcoming the limitations of dry wound dressings. Consequently, modern dressings have been developed with various types, such as films, hydrocolloids, hydrogels, and foams, to address clinical problems [2]. Film dressings are thin, elastic, semipermeable, and often transparent, offering advantages for treating superficial wounds with minimal exudate [7,8]. Hydrocolloid dressings, consisting of an outer semipermeable film and an inner hydrocolloid layer, are widely used for treating both acute and chronic wounds [9]. These dressings can absorb moderate amounts of exudate to create a moist environment, reduce pain, promote angiogenesis and granulation, and inhibit bacterial growth. Hydrogel dressings comprise hydrophilic polymers organized in a three-dimensional network with a high water content, making them suitable for dry or minimally exudating wounds [10]. Foam dressings are a suitable option for chronic wounds with a moderate-to-heavy amount of exudate, as they prevent adherence to the wound bed and support proper epithelialization [8]. However, several cases of contact allergy to foam dressings fabricated with polyurethane have been reported [11]. Foam dressings are commonly fabricated using a foaming technique that involves chemicals such as diphenylmethane diisocyanate, toluene diisocyanate, or diaminodiphenylmethane, which can induce contact allergy [11]. Therefore, there is an unmet need for the development of a biocompatible wound-covering material that can absorb wound exudates while being easy to remove.

Recently, biopolymers such as silk fibroin, agarose, chitosan, collagen, and hyaluronic acid (HA) have been used in the fabrication of wound dressings, aiming to enhance and accelerate the process of wound healing [12,13]. Among them, HA, a primary component of skin extracellular matrix (ECM), is widely used as a raw material in wound dressing fabrication [14]. HA is highly biocompatible and has considerable mechanical strengths that can facilitate skin regeneration by stimulating cell migration, differentiation, and proliferation. Furthermore, HA plays a crucial role in regulating the metabolism and composition of ECM proteins, thereby enhancing their beneficial effects [15]. HA significantly contributes to the complex cascade of events during wound healing [12]. It promotes the recruitment of neutrophils for debris phagocytosis and dead tissue removal, leading to the release of inflammatory cytokines such as tumor necrosis factor-alpha, interleukin-1beta and interleukin-8 [16]. HA, along with fibronectin, guides fibroblast invasion, proliferation, and differentiation into myofibroblasts, essential for collagen deposition and wound contraction [17]. At wound margins, HA interacts with CD44 receptors on keratinocytes, regulating the re-epithelialization process [18]. By harnessing the intrinsic properties and biological functions of HA, diverse wound dressings have been developed [12].

In addition, HA is rich in carboxyl and hydroxyl groups, allowing for the modification of its properties by introducing other functional groups. For instance, HA modified with photocrosslinkable moieties, such as methacrylated HA, exhibited accelerated wound healing efficacy by forming a 3D hydrogel network on the wound area through a photo-induced crosslinking reaction [19,20]. Furthermore, photocurable HAs could be applied as a tissue adhesive, enabling fast wound closure. Given that HA can act as an anti-adhesion barrier [21], it would be beneficial in preventing re-injury upon removal. Therefore, these versatile properties of HA make it suitable for the fabrication of wound dressings.

In this study, we developed a photocrosslikable HA-based biocompatible foam wound dressing to overcome the limitations of conventional wound dressings and minimize the risk of additional damage. To increase mechanical properties and water absorption of the dressing, form-type HA dressings were prepared from hydrogels formed via the photocrosslinking of HA with a high degree of methacrylation (~120%), followed by freeze-drying. After evaluating swelling and lubricant properties, the safety and efficacy of the HA-based wound dressing were assessed through both in vitro and in vivo experiments.

## 2. Materials and Methods

### 2.1. Materials

HA (Mw: 100 kDa) was purchased from SNvia (Busan, Republic of Korea). Methacrylic anhydride (MAA), sodium hydroxide (NaOH), ethanol (purity: 94%), acetone, phenyl-2,4,6-trimethylbenzoylphosphinate (LAP), and potassium sorbate were purchased from Sigma-Aldrich (St. Louis, MO, USA). An epilated porcine skin with 2–3 mm thickness was purchased from a local slaughterhouse.

### 2.2. Synthesis and Characterization of Photocurable HA

Photocurable HA was synthesized as previously reported [22,23]. Briefly, 10.0 g of HA was dissolved in 100 mL of distilled water, and the dissolved HA solution was cooled to 5 °C. Then, a three-fold equivalent of MAA with respect to the disaccharide unit of HA was added dropwise over a 1 h period, with the pH adjusted between 8.0 and 10.0 using 1.0 M NaOH solution. The temperature and pH were maintained for another 23 h. The reaction mixture was precipitated in ethanol, and the precipitated solid was washed with acetone, frozen at −40 °C, lyophilized, and stored at −20 °C until use. ^1^H nuclear magnetic resonance (NMR) spectra were obtained using a Bruker 600-NMR spectrometer (Billerica, MA, USA). The ^1^H NMR spectra were used to confirm the incorporation of methacrylate groups into HA and to calculate the degree of methacrylation (DM).

### 2.3. Preparation of an HA-Based Foam Dressing

To prepare the HA foam used as a wound dressing, a photoinitiator solution was prepared by first dissolving 0.1%(*w*/*w*) LAP in distilled water at 25 °C [23]. Solid methacrylated HA (HAMA) was then added and dissolved for 24 h in the dark to obtain 2%(*w*/*w*) HAMA solution. Potassium sorbate as a preservative was added to the HAMA solution at 0.002%(*w*/*v*). A 5 mL precursor solution was dropped onto a custom-made polypropylene (PP) mold (30 × 30 × 5 mm), photocured under light irradiation at 365 nm with an intensity of 35 mW/cm^2^ (SNvia, Republic of Korea) for 20 s, and freeze-dried. PP molds containing freeze-dried HAMA wound dressings were heat-sealed with an aluminum lid film. The morphology of the cross-sectioned HA foam was observed using scanning electron microscopy (SEM, EM-30N, COXEM, Daejeon, Republic of Korea).

### 2.4. Swelling Test of Photocured HA-Based Foam Dressing

To analyze the swelling behavior of the HA foam (cut to 10 × 10 mm), the HA dressing was weighed and immersed in 20 mL of pH 7.4 PBS in a 100 mm Petri dish at 37 °C for a predetermined time (0, 10, 30, 60, 120, and 240 min). The HA dressing was removed at the indicated timepoints and weighed after the removal of excess moisture from the surface. The swelling ratio, depending on the time, was calculated using the following equation: swelling ratio (%) = (W_t_ − W_0_)/W_0_ × 100, where W_0_ and W_t_ are the initial weight and the weight at the indicated time, respectively [24,25]. Commercial wound dressings (CollaDerm^®^ and CollaHeal^®^ Plus, Hyundai Bioland Co., Ltd., Seoul, Republic of Korea) were used as the controls. Each experiment was repeated three times, and the average values are reported.

### 2.5. Frictional Test of the HA-Based Foam Dressing

Frictional tests were conducted following the modified ASTM standard method (D1894-14) to evaluate the lubricating properties of the HA wound dressings [26]. Porcine skin tissues frozen at −20 °C were thawed at 25 °C to mimic an in vivo frictional circumstance. Porcine skin tissues were cut into a rectangular shape (2.5 cm × 1 cm), and excessive fat layers were trimmed to achieve the same weight (1 g). The porcine skin tissue was covered with soaked gauze to retain moisture until use. The HA foam was cut into a square (2.5 cm × 2.5 cm) and horizontally mounted on a plane using double-sided adhesive tape. Then, the HA foams were hydrated via immersion in PBS for 10 min. Porcine skin tissues were placed lightly and gently on the HA foam and pulled over by a universal testing machine (34 sc-1, Instron, Norwood, MA, USA) in the tensile mode at a uniform speed of 120 mm/min.

### 2.6. Cytotoxicity Study

The three types of wound dressings (HA foam, CollaDerm^®^, and CollaHeal^®^ Plus) were each cut into 5 mg and then sterilized using a UV lamp (0.1 mW/cm^2^, G30T8, Philips, Amsterdam, The Netherlands) for 2 h on both the front and back sides [27]. Subsequently, each dressing was immersed in 1 mL of Dulbecco’s modified Eagle medium supplemented with 10% fetal bovine serum, 100 U/mL of penicillin, and 100 µg/mL of streptomycin for 48 h. Normal human dermal fibroblasts (nHDFs) were seeded in a 24-well plate at 5 × 10^4^ cells/mL and cultured in a 5% CO_2_ incubator at 37 °C. Following a 24 h incubation period, the cells were exposed to the extracts from each wound dressing. Medical gauze was used as a control. After 24 h, the 3-(4,5-dimethylthiazol-2-yl)-2,5-diphenyl tetrazolium bromide (MTT) assay was performed as per our previous protocol [28]. Ext-racts and media were removed, and the cells were incubated in 0.5 mg/mL of MTT solution for 4 h in a 5% CO_2_ incubator at 37 °C. Subsequently, the MTT solution was discarded, and 300 µL of dimethyl sulfoxide was added to dissolve the formazan crystals. The reduction in the MTT reagent was then quantified by measuring the absorbance at 570 nm using an ELx800 absorbance microplate reader (BioTek Instruments, Inc., Winooski, VT, USA).

### 2.7. Cell Adhesion Study

Three types of wound dressings, HA form, CollaDerm^®^, and CollaHeal^®^ Plus, were cut into circles with a 1.5 cm diameter and UV-sterilized for 2 h on both the front and back sides. Next, 200 μL of Matrigel at 5 mg/mL was dispensed into 24-well plates and incubated in a 5% CO_2_ incubator at 37 °C for 20 min. Subsequently, each wound dressing was placed on Matrigel, and nHDFs were seeded onto the wound dressings at 1 × 10^5^ cells/mL. Cells were cultured at 5% CO_2_ and 37 °C. After 24 h, the culture media were removed, and 1 mL of 4% paraformaldehyde was added for fixation at 44 °C for 4 h, followed by sequential dehydrated in ethanol solutions: 10%, 30%, 50%, 70%, and 100%; each dehydration stage was conducted at room temperature for 10 min. Finally, the wound dressings and cells were dried at 37 °C overnight. The morphology of cells attached to the surface of the wound dressing was confirmed via SEM.

### 2.8. Animal Experiments

The protocol for the animal experiment conducted in this study was reviewed and approved by the Pusan National University Institutional Animal Care and Use Committee (PNU-IACUC; Approval Number PNU-2021-0087). All rats were managed at the Pusan National University Laboratory Animal Resources Center, which is accredited by the Ministry of Food and Drug Safety. Seven-week-old male Sprague–Dawley (SD) rats (*n* = 36) were procured from Samtako Bio Korea (Osan, Republic of Korea). Rats were housed in standard cages, provided with basal feed, and maintained on a 12 h light/dark cycle. A minimum acclimatization period of one week was observed before the experiment began. The animal room was maintained at a temperature of 23 ± 2 °C with appropriate humidity.

Following acclimatization, the 36 rats were divided (9 rats per group) into four groups: gauze, HA form, CollaDerm^®^, and CollaHeal^®^ Plus. The initial body weight was recorded. Anesthesia was induced in rats using a mixture of Zoletil (Virbac SA, Carros, France) and Rompun (Bayer Korea, Seoul, Republic of Korea) solution at a 2:1 ratio (1 mg/kg intramuscular injection). A 10 × 10 cm^2^ area of the dermis was shaved using an electric shaver, and 5 mg/kg of carprofen was administered subcutaneously as an analgesic. The dorsal skin was sterilized with povidone and 70% ethanol, and a 2 × 2 cm^2^ section of the skin was excised to create a full-thickness defect. After capturing an image of the affected area, the wound dressing was applied and secured firmly to the rat using medical gauze and tape. Sulfamethoxazole (1 mg/mL) and trimethoprim (0.2 mg/mL) were added to the drinking water. The wound dressing was changed every 4 days, and during each change, a photograph of the affected area was taken, and the weight was measured. On day 24, rats were euthanized using CO_2_, and tissue samples were collected. Livers and kidneys were extracted, rinsed with PBS, and weighed. The dermis of the defect was removed, rinsed with PBS, and used for protein analysis and histological examination.

### 2.9. Histological Analysis

A histological analysis of the skin tissue was conducted following our previously established protocol [29]. Skin tissue samples were fixed in 4% neutral buffered formaldehyde (pH 6.8) for 24 h. Subsequently, each tissue sample was dehydrated and embedded in paraffin. Skin sections (4 μm thick) were then obtained from the paraffin-embedded tissue using a Leica microtome (Leica Microsystems, Bannockburn, IL, USA). The sections were subsequently deparaffinized using xylene and rehydrated with sequentially decreasing ethanol concentrations ranging from 100% to 70%, followed by washing in distilled water. Skin sections were stained with hematoxylin and eosin (H&E) (Biognost, Zagreb, Croatia), followed by washing with dH_2_O. The extent of the affected area was determined using the Leica Application Suite (Leica Microsystems, Bannockburn, IL, USA).

### 2.10. Western Blotting

Proteins from the tissue samples were isolated using Pro-prep solution (iNtRON Biotechnology Co., Seoul, Republic of Korea) according to the manufacturer’s protocol. The extracted proteins were separated via 8% sodium dodecyl sulfate-polyacrylamide gel electrophoresis and then transferred to nitrocellulose membranes (Daeil Lab Service Co., Ltd., Seoul, Republic of Korea). The membranes were blocked with 5% skimmed milk (Difco, Sparks, MD, USA) in PBS containing 0.05% Tween-20 (PBST). After a 2 h incubation period, the membranes were treated with the following antibodies: mouse anticollagen type 1A1 (1:500; cat no. sc-293182, Santa Cruz Biotechnology, Inc., Santa Cruz, CA, USA), rabbit anticollagen type 1A2 (1:1000; cat no. ab96723, Abcam, Cambridge, UK), mouse anticollagen type 3A1 (1:500; cat no. sc-271249, Santa Cruz Biotechnology), rabbit antielastin (1:1000; cat no. ab217356, Abcam), rabbit anti-MMP-1 (1:1000; cat no. ab137332, Abcam), and rabbit anti-COX-2 (1:1000, cat no. S4842, Cell Signaling Technology Inc., Danvers, MA, USA) overnight. Blots were incubated with horseradish peroxidase-conjugated anti-rabbit or anti-mouse antibodies (diluted 1:2000 in 5% skimmed milk with PBST) for 1 h. Luminol (Bio-Rad, Hercules, CA, USA) was used to visualize antibody binding. The membranes were subsequently probed with antibodies against β-actin (1:5000; cat. no. S4967, Cell Signaling Technology Inc.), which was used as an internal control. The blots were scanned using Gel Doc 1000 version 1.5 (Bio-Rad), and the protein band intensities were normalized to the GAPDH band intensity.

### 2.11. Statistical Analysis

The results are presented as the mean ± SEM. Data were subjected to one-way analysis of variance (ANOVA; SPSS for Windows, 10.10, standard version; SPSS Inc., Chicago, IL, USA). Means derived from three independent experiments were statistically evaluated using analysis of variance along with Duncan’s multiple range test. The significance was established at a threshold of *p* < 0.05.

## 3. Results and Discussion

### 3.1. Synthesis and Characterization of HA Functionalized with Photocurable Groups

HA plays a major role in wound-healing processes [18], and HA-based wound dressings are potential candidates for wound-healing applications with moderate-to-heavy exudates and high water absorption [12]. However, strategies to improve the mechanical stability of HA-based wound dressings are required. HA can be easily modified because of the presence of carboxy and hydroxyl groups within the repeating polymer chains [22,30]. Herein, we induced the rapid and precise crosslinking of HA while enhancing the mechanical stability by functionalizing HA with the photocurable group MA. The primary hydroxyl groups in the HA were trans-esterified and incorporated with MA groups under aqueous basic conditions (Figure 1A). After the synthesis of HAMA, ^1^H NMR analysis was conducted to calculate the DM of HAMA chains [22]. The DM was calculated from the relative integration of the methacrylate protons (5.6 and 6.0 ppm, indicated as 1 and 2, respectively, in Figure 1B) to the methyl protons in HA (1.9 ppm, indicated by 5 in Figure 1B), and the DM value was approximately 120%, which is suitable for rapid photocuring.

### 3.2. Preparation and Characterization of the HA-Based Foam Dressing

HA foam was fabricated via freeze-drying using a high molecular weight HA of 1000–4000 kDa, which had been chemically crosslinked or mixed with additional materials such as chitosan, alginate, dextran, and carboxymethyl cellulose to improve the mechanical stability [31,32]. To enhance mechanical stability, the HA-based wound dressings were manufactured through the following continuous process to facilitate mass production. A precursor solution containing the HAMA polymer, photoinitiator, and preservative was placed into a PP mold and photocured for 30 s using a custom-made light irradiator. The photocured HAMA hydrogels were freeze-dried and stored at room temperature after heat-sealing (Figure 2A). The freeze-dried HA-based wound dressing was made of a white-colored sponge (Figure 2B). The SEM image of the cross-sectioned sample showed a highly porous structure with an average pore size ranging from 50 to 300 μm. The time-dependent swelling behavior of wound dressings in PBS buffer for 6 h is shown in Figure 2C. Equilibrium swelling states for all wound dressings were reached within 10 min. The CollaDerm^®^ and the HA foam exhibited a higher swelling ratio (~4000%) than that of CollaHeal^®^ Plus (~2000%). HA foam with its porous structure is suitable for the high absorption of exudate from the wound [33].

Since excessive friction created by applied dressings can disrupt the healing process by causing additional damage to the wound area [34], dressings with low friction properties are considered desirable. To investigate the lubricating properties, the frictional force between the porcine skin tissue and the hydrated wound dressing was measured via mechanical stimulation (Figure 2D). CollaDerm^®^, commonly used in surgery, was used as a control, whereas CollaHeal^®^ Plus was not available to conduct a test because of its dissolution. The maximum frictional force of CollaDerm^®^ was 85.38 ± 13.2 mN, while that of the HA foam was 44.85 ± 5.93 mN, which was half that of the control (Figure 2F). HA and HA derivatives have been used to reduce postsurgical adhesion based on their antiadhesive properties [35]. The lubricating efficacy of HA foam minimizes inflammation induced by mechanical stimulus at the contact interface of the lesion.

### 3.3. Assess the Cytotoxicity of Wound Dressings on nHDFs

To determine the cytotoxicity of the manufactured wound dressings, nHDFs were exposed to the extract from each wound dressing, followed by the MTT assay (Figure 3A(i)). When the wound dressing is applied to the wound, the dressing material may dissolve in the wound exudate, potentially affecting the wound [36,37]. If the extract from the wound dressing material had a detrimental effect on the cells, this could significantly reduce cell viability [38]. However, no significant alteration in cell viability across all groups compared with the no-treatment group was observed. Consequently, we conclude that the wound dressing materials did not exhibit cytotoxicity toward nHDFs (Figure 3B).

### 3.4. Evaluation of Adhesion of nHDFs to Wound Dressings

Wound dressings applied to injured skin must be periodically replaced to prevent inflammation. However, the removal of wound dressings can potentially cause re-injury if cells adhere tightly to the dressing [39]. Therefore, it is advantageous to avoid this. Consequently, we used SEM to analyze the morphology of nHDFs after culturing them on the wound dressing (Figure 3A(ii)). In general, cells secrete adhesive proteins to establish interactions with a surface, thereby promoting cell spreading. Consequently, upon adhering to a surface, cells are unable to both spread and maintain a spherical shape [40,41]. Thus, cells that successfully attach to the surface appear larger, while those that do not adhere remain smaller [41,42,43]. Herein, the length of cells cultured on gauze and CollaHeal^®^ Plus was approximately 36.6 and 25.8 nm, respectively, while the length of cells cultured on HA foam and CollaDerm^®^ was approximately 18 and 11.6 nm, respectively (Figure 3C). Therefore, skin cells adhered to the gauze and CollaHeal^®^ Plus but did not strongly adhere to HA foam and CollaDerm^®^.

### 3.5. Effects of Wound Dressings on the Skin of Rats with Full-Thickness Wound Defects

To assess the wound-healing capacity of the wound dressing in vivo, we induced full-thickness excisional wounds in the dermis of rats and treated them with the wound dressing for 24 days. As illustrated in the schedule in Figure 4A, wound dressings were replaced every 4 days, and wound-healing progress was monitored through photographs. HA foam and CollaDerm^®^ exhibited faster observable wound healing than gauze and CollaHeal^®^ Plus (Figure 4B). Notably, rats with gauze and CollaHeal^®^ Plus experienced re-injury and bleeding during dressing changes up to the 8th day, whereas the rats with HA foam and CollaDerm^®^ had minimal additional bleeding. This result occurred because the gauze and CollaHeal^®^ Plus created a dry wound environment, clumped at the wound site, so that when the dressing was removed, the incompletely organized scab was removed as well. On the other hand, HA foam and CollaDerm^®^ allowed the wound to recover in an appropriately moist environment, did not stick to the wound area, and prevented the wound scab from falling off along with the dressing when the dressing was removed. Importantly, this bleeding was not attributed to the toxicity of the wound dressing, and this was substantiated by the absence of significant changes in the liver, kidney, and body weight of the rats following wound dressing treatment. Bleeding that occurred up to the 8th day was considered secondary damage during dressing changes.

The wound healing ratio was calculated as the reduced area compared with the initial wound area on day 0 and is depicted in Figure 4E. No significant difference between gauze, HA foam, and CollaDerm^®^ were evident on the 8th day. However, on the 16th day, the HA foam achieved a wound healing ratio of 91.3%, whereas that of CollaDerm^®^, CollaHeal^®^ Plus, and gauze was 81.3%, 79.3%, and 77.7%, respectively (Figure 4C). By day 24, the HA foam demonstrated a wound healing ratio of 98.8%, whereas this was 97.8%, 90.4%, and 90.1% for CollaDerm^®^, CollaHeal^®^ Plus, and gauze, respectively. In comparison with other wound dressings, the HA foam displayed the fastest and highest wound healing rate. However, these results were only based on the photographic recovery of skin wounds. Therefore, we further verified the healing process of the internal skin.

### 3.6. Histological Analysis of the Skin Surface of Rats with Full-Thickness Defect Wounds

To confirm whether the damaged skin had fully recovered, we conducted a histological analysis using H&E staining. We employed two major key indicators to test the restoration of damaged skin. First, a matured and restored epidermis generally forms uniformly and appears flat, and second, matured granulation tissue forms in the dermis [44,45]. The experimental results indicated an injured area in the epidermis (indicated by arrows) remained with gauze, CollaDerm^®^, and CollaHeal^®^ Plus and that the thickness of the epidermis was not uniform, indicating that the epidermis had not fully matured (Figure 5A). Conversely, the HA foam displayed a matured epidermis with uniform thickness and a flat surface. In addition, gauze, HA form, and CollaDerm^®^ demonstrated the presence of matured granulation tissue, whereas CollaHeal^®^ Plus only exhibited the initial stages of granulation tissue. Consequently, the group treated with the HA foam exhibited the most matured skin structure.

### 3.7. Effects of Wound Dressings on the Expression of ECM Proteins

As the skin undergoes recovery, the active functioning of dermal fibroblasts increases the expression of ECM proteins, such as collagen and elastin, whereas the expression of MMP-1, an enzyme responsible for ECM protein degradation, decreases [46]. We confirmed these cellular microenvironments via Western blotting. The expression of COL1A1 was significantly higher in the HA foam, CollaDerm^®^, and CollaHeal^®^ Plus groups than in the gauze group, and the expression of COL1A2 and COL3A1 was significantly higher in the HA foam and CollaDerm^®^ groups (Figure 5B–E). Additionally, the HA foam group tended to increase the expression of elastin compared to that in the gauze group (Figure 5F). However, MMP-1 expression did not change (Figure 5G). To assess the potential risk of the wound dressing causing inflammation in the damaged skin, we confirmed the expression of COX-2, a representative inflammatory factor whose levels rise in response to inflammation induction. Given that no significant alteration occurred in the expression of COX-2, we concluded that none of the tested wound dressings induced inflammation in the rat skin (Figure 5H).

## 4. Conclusions

HA is widely recognized for its natural material properties, including high biocompatibility and nonimmunogenicity. However, as a hydrophilic material, HA easily adheres to the skin and is unsuitable for use as a wound dressing [12]. Conversely, when HAMA is photopolymerized and freeze-dried, the advantages of HA are retained, and this becomes more suitable as a wound-dressing material because it does not dissolve in bodily fluids or stick to the skin. In this study, HA from wound dressings did not induce toxicity or inflammation in skin cells and rats. Additionally, HA foam did not cause re-injury when replacing wound dressings because of their low adhesive property to skin cells compared with that of gauze and other wound dressings. Consequently, the HA foam wound dressings exhibited rapid and effective wound healing. This effect was not limited to epidermal wounds; dermal wounds also achieved complete recovery, characterized by the formation of mature granulation tissue and increased production of ECM proteins, particularly collagen. Therefore, HA foam wound dressings prepared by photopolymerizing and freeze-drying HAMA can be used as high-performance wound dressings that facilitate rapid wound healing without causing re-injury. However, the findings of this study have limitations, as it was conducted on only SD rat skin. While the skin of SD rats is often used in wound healing experiments due to its structural and thickness similarities to human skin, there are differences in the composition of ECM substances and immune responses compared to human skin. Therefore, further research conducted on human skin is necessary.

## Figures and Tables

**Figure 1 biomedicines-12-00510-f001:**
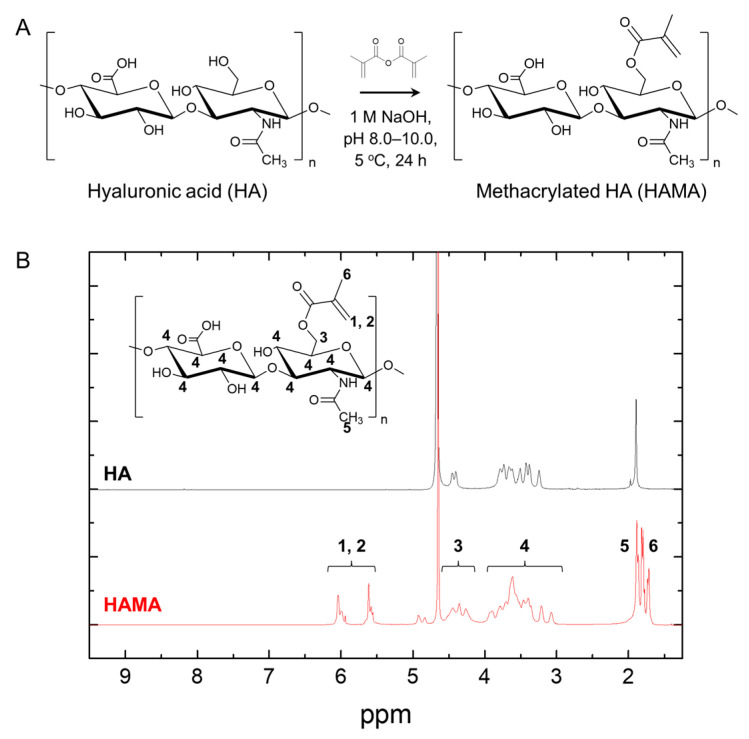
(**A**) Scheme for preparing methacylated hyaluronic acid (HAMA) and (**B**) representative ^1^H NMR spectrum of HA and HAMA. Peaks correspond to methacrylate protons (1 and 2) and methyl protons (5) in the HAMA chains.

**Figure 2 biomedicines-12-00510-f002:**
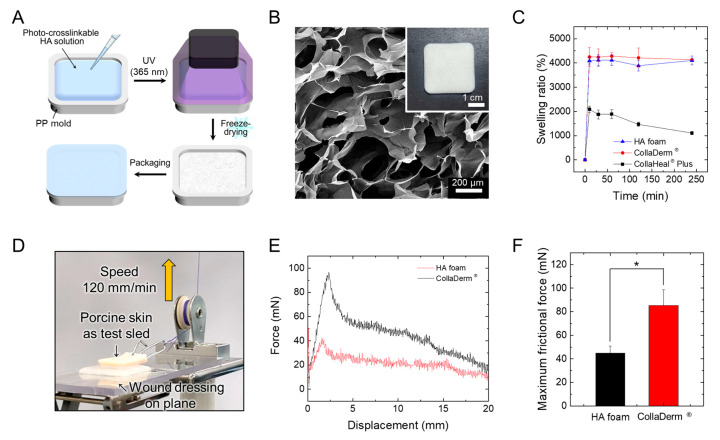
(**A**) Schematic showing the fabrication procedure of a photocured HA-based wound dressing. (**B**) Photo and SEM images of the photocured HA-based wound dressing. (**C**) Time-dependent swelling ratio of wound dressings after immersion into PBS (pH 7.4) at 37 °C (*n* = 3). (**D**) Photograph showing the experimental set-up for frictional force measurement of wound dressing. (**E**) Force-displacement profiles obtained from friction tests on porcine skin tissue using hydrated wound dressings. (**F**) Comparison of the maximum frictional force between the commercial wound dressing (CollaDerm^®^) (*n* = 3). The asterisk (*) indicates statistical significance with *p* < 0.05.

**Figure 3 biomedicines-12-00510-f003:**
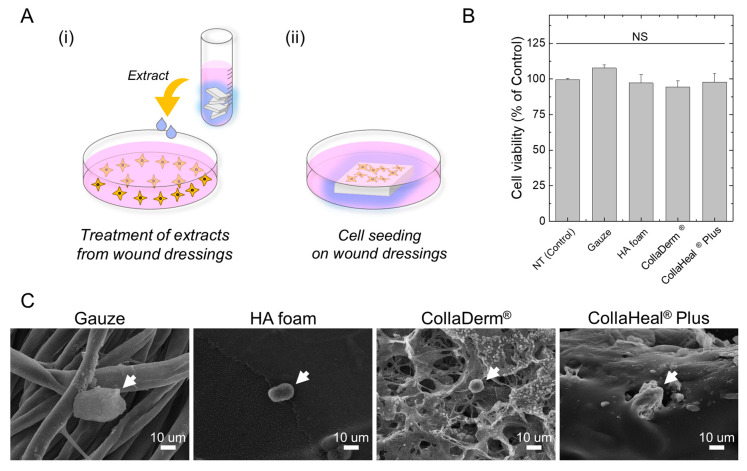
(**A**) Schematic of the (i) cytotoxicity and (ii) cell adhesion tests. (**B**) Cell viability of nHDFs treated with wound-dressing extract (*n* = 3). (**C**) SEM images of nHDFs cultured on wound dressing with Matrigel. The arrow indicates nHDFs. NS indicates not significant (*p* > 0.05).

**Figure 4 biomedicines-12-00510-f004:**
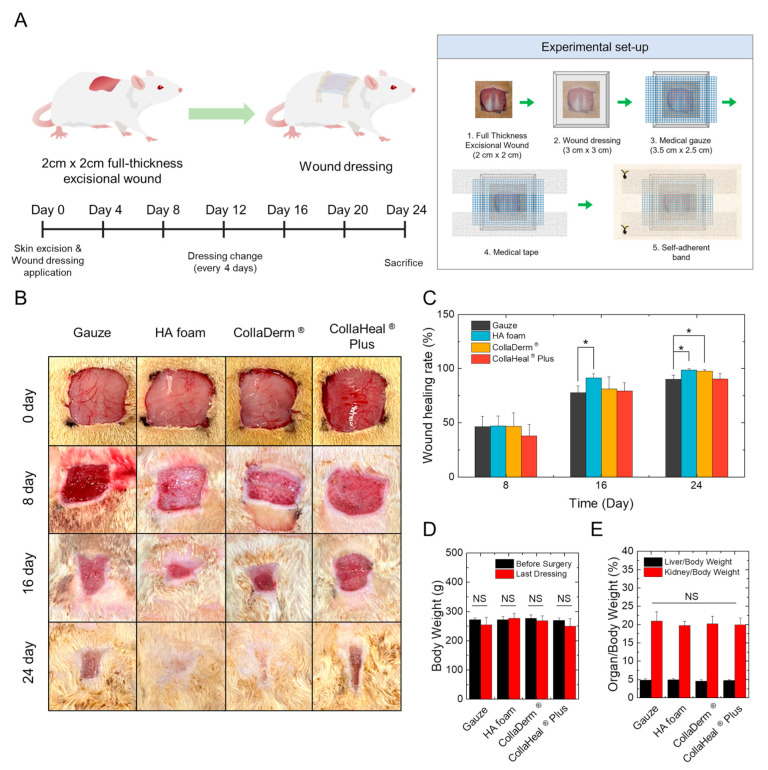
(**A**) Schematic and timeline of in vivo surgery to induce a full-thickness excisional wound. (**B**) Wound images of rats treated with wound dressings for 24 days after the induction of full-thickness skin defects and (**C**) graph showing the wound healing rate (*n* = 9). Changes in (**D**) body weight and (**E**) organ weight relative to body weight in rats treated with wound dressings. * *p* < 0.05 compared with the gauze group. NS indicates not significant (*p* > 0.05).

**Figure 5 biomedicines-12-00510-f005:**
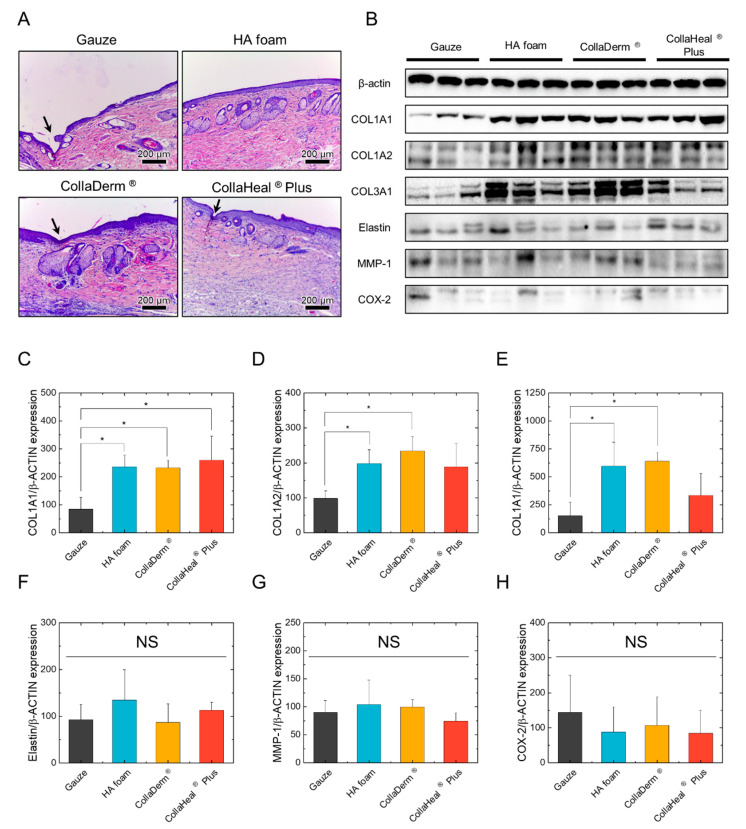
(**A**) Histology of rat wounds analyzed via H&E staining (*n* = 9). The arrow indicates the immature epidermis. Scale bar: 200 μm. (**B**) Expression of proteins extracted from rat dermal skin with wounds for (**C**) COL1A2, (**D**) COL1A2, (**E**) COL3A1, (**F**) elastin, (**G**) MMP-1, and (**H**) COX-2 (*n* = 9). * *p* < 0.05 compared with the gauze group. NS indicates not significant (*p* > 0.05).

## Data Availability

The datasets used and/or analyzed in the current study are available from the corresponding author on reasonable request.

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
