# Peer review of "Antiadhesive Hyaluronic Acid-Based Wound Dressings Promote Wound Healing by Preventing Re-Injury: An In Vivo Investigation"

_biomedicines, 2024, doi:10.3390/biomedicines12030510_

Round 1

Reviewer 1 Report

Comments and Suggestions for Authors

In this manuscript, the authors propose an antiadhesive wound dressing. They claimed a promising biocompatible foam wound dressing based on modified hyaluronic acid, which offers enhanced wound-healing capabilities. The research is well planned, presented, and organized.

However, some improvements that could have been value-added are:

1.       The introduction needs to be reviewed more by including:

2.      Formulation of methacylated HA-based hydrogels

3.      Wound-covering material based on HA

4.      The major references are obsolete and should be updated with more recent and relevant ones.

5.      For the UV sterilization, the authors should identify the device of the UV light source used for that. As well as the concertation of exposition (mW/cm2)

6.      Graphs in figures should indicate clearly in the legend if there is no significant difference compared to the controls (Fig. 3 B, Fig. 4 DE, Fig. 5 FGH, etc.).

7.      Graphs in figures should indicate clearly in the legend the significant difference compared to controls (Fig. 2 F, Fig. 4 C, etc.).

8.      The conclusion should be more informative and needs an outlook, as well as the limitations of this study.

Author Response

Thank you for your review of our paper. Please see the attachment.

Reviewer 2 Report

Comments and Suggestions for Authors

The authors have executed this in vivo investigation on 36 rat models to assess the biocompatibility and wound-healing capabilities of 4 different wound-dressing materials. I find the study questions of this paper to be in line with the trend of developing the most biocompatible wound-dressings possible. The study structure and methodologies are all properly detailed and discussed. However I have some concerns and questions regarding their introduction and methodologies which I have listed them all down below:

Title:

Even though the title of this paper in compelling and easy to comprehend, I highly encourage the authors to include the type of their paper in their title: 

“Antiadhesive hyaluronic acid-based wound dressings promote wound healing by preventing re-injury: An in vivo investigation”

Abstract:

The overall structure of the abstract is appropriately executed. However, there are missing data regarding the type of animal model, the total number of models assessed in vivo, the evaluation periods, the kind of injury (e.g., burning, stabbing, etc.) and the size of the injury. Kindly include these data in your abstract. 

Keywords:

I highly suggest authors add these terms to their list of keywords: “wound dressings” and “biocompatibility”.

Introduction:

“Consequently, modern dressings have been developed using various products such as films, hydrocolloids, hydrogels, and foams to solve clinical problems [7].”

I appreciate the authors efforts to refer to reference number 7 as their support for this claim. And I am pretty sure that this statement may probably not need a revision, however the fact that such important statement in the beginning of your introduction is supported by a 5-years old reference is simply not ideal and significantly tarnishes the reliability of your claims.

I highly suggest the authors extend their electronic search on the online databases and try to include the newest study possible for each of their statements and claims that are not of their own or their own opinion. Doing so, the readers will have no other choice but to trust your position and your research question since you have executed your paper based on only the newest published studies and review articles.

“Herein, a biocompatible foam wound dressing was developed to facilitate efficient wound recovery without the risk of further injury. To manufacture an HA-based foam dressing, the hydroxyl groups of the HA were modified to methacrylate groups, enabling rapid photocuring. The photocured HA solution was freeze-dried to create porous structures for high absorption of exudate. The HA-based foam dressing (HA foam) exhibited high water absorption and antifriction properties compared with conventional biopolymer-based foam dressings and was nontoxic in vitro. Moreover, the HA foam showed favorable wound-healing properties without promoting an inflammatory reaction in vivo by reducing secondary damage during dressing changes.”

The last paragraph of the introduction of an original paper must be comprised of only the main goals of the paper and maybe even a couple of hypotheses or study questions to be answered. However, in this paper the authors have chosen to include the main conclusions of their study in this section of their introduction. In my opinion, this is absolutely unnecessary and makes the rest of the study seem pointless. Firstly, you have reported the main conclusions of your research in your abstract. Now at the end of your introduction, you should only stick to your study questions, the research gap that you have found in your extensive search in the literature, and your main goals and purposes with this study. Nothing more and nothing less. Kindly revise your introduction to meet the standards of all proper original studies. 

Materials and methods:

2.1. materials

“Porcine skin was purchased from a local slaughterhouse.” I appreciate the authors honesty and transparency; however, it would have been much better if the authors could include a little bit more details regarding the porcine skin that they have used in their experiments for the sake of the readers better understanding your choices.

Throughout your methods and materials section, I did not encounter may references to other original studies. Is that because most of your study steps were only developed by your team? Cause if that is not the case, I highly recommend citing the references that the authors used to develop and follow their study instructions.

2.8. Animal Experiments

“the 36 rats were divided (nine rats per group) into four groups: Gauze, HA form, CollaDerm®, and CollaHeal® Plus.” The detailed data regarding the 4 different study groups and the total number of animal models must definitely be included in your abstract as I mentioned before. 

“and a 2 × 2 cm2 section of the skin was excised to create a full-thickness defect.” 

As I mentioned before, the size and type of the skin defect must also be included in the abstract. These are key information that can replace some not so important background statements in your abstract.

“On day 24, rats were euthanized using CO2 and tissue samples were” the evaluation period of 24 days must be included in the abstract.

Author Response

(The authors gave the same response as above.)

Reviewer 3 Report

Comments and Suggestions for Authors

The manuscript presents a comprehensive study on the development and characterization of hyaluronic acid-based foam wound dressings (HA foam). The authors claimed the enhanced wound-healing capabilities of HA foam due to its  antifriction properties, and low adhesiveness, which minimizes re-injury during dressing changes. Overall, I find this work to be of significant interest for the field of regenerafive medicine. However, there are several points that need to be addressed before the manuscript can be considered for publicafion.

- On Page 9, section 3.5, the results depicting the wound-healing ratio in Figure 5C require analysis using appropriate statistical testing to validate the findings.

- The manuscript should clarify the relevance and impact of friction properties on the wound healing process, as the current reasoning remains unclear.

Minor points:

- The captions of relevant figures need to include sample sizes to provide a clearer context and understanding of the experimental design and results.

- In Figure 5A, the scale bar noted in the caption should be corrected to 200 µm.

Comments on the Quality of English Language

The English writing is generally clear and coherent. However, it does contain a few typos that need correction.

Author Response

(The authors gave the same response as above.)

Round 2

Reviewer 1 Report

Comments and Suggestions for Authors

The authors addressed all the reviewers' concerns. In my opinion, the revised paper is now acceptable.

Author Response

Thank you for your kind review of our paper. Please see the attachment.

Reviewer 2 Report

Comments and Suggestions for Authors

In my initial review of this paper, I asked the authors to revise their title to be more specific and clear to the readers. The authors have revised their title accordingly.

There was missing data regarding the methodology of this paper in their abstract. There was a lot of information regarding the animal models, their wounds, their groups, etc. missing from the abstract. I asked the authors to revise their abstract to include these key data and the authors have done accordingly.

I asked the authors to add two new terms to their list of keywords and they have added them accordingly.

There were some references in the introduction that simply did not do justice to the authors' claims due to their old publication date. I asked the authors to update their electronic search to include more recently-published studies as their references and they have done so accordingly.

Regarding the last paragraph of their introduction, the authors initially showcased their main results and conclusions. I asked the authors to remove the conclusions from their introduction and dedicate the last paragraph of their introduction solely to their main goals and purposes with this study and if there are any hypotheses along their study questions. The authors have revised their introduction accordingly.

Regarding the porcine skin that was used in this study, I asked the authors to provide more details on the source of their investigated skin which the authors have done so accordingly.

In different methodology sections of this paper, the authors detailed many different steps and protocols without referring to other similar studies. I asked the authors about the originality of these protocols and if they were following other studies' methodologies to refer to them in each section of their methods. The authors have provided multiple references for each section and have clearly stated which study was followed for each of the steps of their investigations.

Overall, I believe the authors have revised their manuscript according to all of my concerns and suggestions. However, I encountered a couple of English grammatical errors and I kindly ask the authors to fix those issues immediately.

Comments on the Quality of English Language

Overall, I believe the authors have revised their manuscript according to all of my concerns and suggestions. However, I encountered a couple of English grammatical errors and I kindly ask the authors to fix those issues immediately. 

Author Response

(The authors gave the same response as above.)
